# The Influence of Diabetes on Multisensory Integration and Mobility in Aging

**DOI:** 10.3390/brainsci11030285

**Published:** 2021-02-25

**Authors:** Jeannette R. Mahoney, Joe Verghese, Claudene George

**Affiliations:** 1Department of Neurology, Division of Cognitive & Motor Aging, Albert Einstein College of Medicine, Bronx, NY 10461, USA; jeannette.mahoney@einsteinmed.org (J.R.M.); clgeorge@montefiore.org (C.G.); 2Department of Medicine, Division of Geriatrics, Montefiore Medical Center, Bronx, NY 10467, USA

**Keywords:** aging, multisensory integration, sensorimotor integration, mobility

## Abstract

(1) Background: one out of every four adults over the age of 65 are living with diabetes, and this alarming rate continues to increase with age. Diabetes in older adults is associated with many adverse health outcomes, including sensory and motor impairments. The objective of this exploratory study was to determine whether diabetes influences the interplay between multisensory integration processes and mobility in aging. (2) Methods: in this cross-sectional observational study, we recruited 339 non-demented older adults (76.59 ± 6.21 years; 52% female, 18% with diabetes). Participants completed a simple reaction time test in response to visual, somatosensory, and combined visual-somatosensory stimulation. Magnitude of visual-somatosensory integration was computed and served as the independent variable. (3) Results: logistic regression revealed that presence of diabetes was inversely associated with the magnitude of visual-somatosensory integration (β = −3.21; *p* < 0.01). Further, mediation models revealed that presence of diabetes negatively influenced the relationship of visual–somatosensory integration magnitude with balance (95% CI −0.16, −0.01) and gait (95% CI −0.09, −0.01). Participants with diabetes and taking insulin (*n* = 14) failed to integrate sensory information entirely; (4) conclusions: taken together, results from this exploration provide compelling evidence to support the adverse effect of diabetes on both multisensory and motor functioning in older adults.

## 1. Introduction

More than 100 million adults are living with diabetes or pre-diabetes in the United States; an alarming rate that continues to increase with chronological age. According to the Centers for Disease Control and Prevention, over 25% of individuals age 65 years and older are diagnosed with diabetes [1]. The American Diabetes Association highlights that diabetes in older adults is associated with many adverse health outcomes, including increased rates of mortality, functional disability, vascular co-morbidities (including heart disease and stroke), as well as greater risk for polypharmacy, cognitive impairment, pain, and injurious falls [2]. Diabetes and insulin resistance are notorious for their contribution to an accelerated aging process [3].

It is well known that healthy aging presents many challenges to the central nervous system that concurrently disrupt cognitive, sensory, and motor functionality [4]; suggesting a common underlying neural mechanism [5,6]. In a recent multisensory study of 345 older adults, we demonstrate that cognitive impairment influences visual–somatosensory integration, which negatively impacts balance and gait [7]. There we posit that sensory, cognitive, and motor processes rely on overlapping neural circuitry involving prefrontal cortex (PFC) connections with other cortical, subcortical, cerebellar regions that are compromised in older adults with cognitive impairments. However, little research has focused on investigating diabetes-related alterations of multisensory functioning on cognitive or motor performance. There is good reason to believe that diabetes would be associated with similar declines in magnitude of multisensory integration as seen in Alzheimer’s disease, since both disease processes are linked to changes in cerebrovascular function, oxidative stress, advanced glycation end products (AGEs), and insulin signaling system impairments [8]. A *Nature Reviews* neurology study, conducted by Sims-Robinson et al., highlights the idea that neurodegeneration and cognitive impairments in type 2 diabetes could be related to interruptions in insulin receptor signaling, which can lead to increased accumulation of amyloid-β and tau depositions—significant biomarkers for Alzheimer’s disease [9].

Work from our lab and others has linked diabetes to alterations in cognitive (namely executive functions) and motor outcomes in older adults (quantitative gait) [10,11]. These findings provide evidence that diabetes attenuates neural responses in the prefrontal cortex (PFC). Other work in middle-aged adults (*n* = 375; age 18–64) with and without type 1 diabetes reveals that, while diabetes was not associated with declines in problem-solving or learning and memory, it was associated with reductions in psychomotor speed as measured by performance on the Grooved Pegboard [8]. Collectively, these findings indicate that disease-related reductions in cognitive and motor functionality could onset at varying time points, but nevertheless are likely associated with disease-related alterations in PFC activation.

Given the link between diabetes, decreased PFC activation, and decreased cognitive and motor abilities, as well as the notion that PFC regulates multisensory, cognitive, and motor processes, the current study was specifically designed to determine: 1) if diabetes in older adults is associated with decreased magnitude of visual–somatosensory integration, and 2) whether diabetes influences the relationship of visual–somatosensory integration with balance and gait.

## 2. Materials and Methods

Four-hundred-thirteen participants enrolled in the Central Control of Mobility in Aging (CCMA) longitudinal study or the Visual-Somatosensory Integration (VSI) study at the Albert Einstein College of Medicine (AECOM), New York (USA), completed a simple multisensory reaction time (RT) experiment between June 2011 and January 2019. CCMA and VSI study eligibility criteria required that participants be 65 years of age and older, reside in lower Westchester county, and speak English. Exclusion criteria at enrollment included inability to independently ambulate, dementia diagnosis, significant bilateral vision and/or hearing loss, active neurological, or psychiatric disorders that would interfere with evaluations, recent or anticipated medical procedures that would affect mobility, and/or receiving hemodialysis treatment [12,13].

All participants were required to have bilateral visual acuity that was better or equal to 20/100 as measured by the Snellen eye chart. Individuals who were unable to hear a 2000 Hz tone at 25 dB in both ears were not included. Presence or absence of neuropathy was diagnosed by study clinicians following a standard neurological examination, and participants with severe neuropathy (unable to feel somatosensory stimulation) were not included. Additional exclusion criteria for the current study included: prevalent dementia (*n* = 12); Parkinson’s disease (*n* = 3); presence of severe visual impairments (e.g., glaucoma, macular degeneration, detached retina, and monocular blindness *n* = 12); deafness (*n* = 1); and/or inadequate multisensory behavioral performance (*n* = 46).

After exclusions, the overall study cohort consisted of 339 older adults (mean age 76.59 ± 6.21 years; 52% female). All participants provided written informed consent to the experimental procedures, which were in accordance with the Declaration of Helsinki and approved by the institutional review board of the Albert Einstein College of Medicine (IRB number: 2016-6936; IRB approval date: 02/07/2020).

### 2.1. Experimental Design

Participants completed a simple reaction time paradigm employing three sensory conditions that were presented bilaterally: two unisensory (visual and somatosensory) and one multisensory (simultaneous visual-somatosensory). Specific details regarding the multisensory experimental protocol, equipment, and data processing procedures are available [14]. In short, individuals were seated and instructed to quickly press a foot pedal as soon as they saw or felt stimulation from the multisensory apparatus (see also Figure 1 adapted from [15]). Visual and somatosensory stimuli were delivered through a custom-built stimulus generator (Zenometrics, LLC; Peekskill, NY, USA) that consisted of two control boxes, each housing a 15.88 cm diameter blue light emitting diodes (LEDs) and a 30.48 mm × 20.32 mm × 12.70 mm plastic housing containing a vibrator motor with 0.8 G vibration amplitude. The devices were connected to a network control center, which allowed direct control for each device through the testing computer’s parallel port. The devices were cycled on and off at precise predetermined intervals. A TTL (transistor-transistor-logic, 5 V, duration 100 ms) pulse was used to trigger the visual and somatosensory stimuli through E-Prime 2.0 professional software.

Control boxes were also mounted to the apparatus, which participants rested their hands upon comfortably, with index fingers placed over the vibratory motors on the back of the box and their thumb on the front of the box, under the LED. A third dummy control box was placed in the center of the actual control boxes, at an equidistant length (28 cm), and contained a bull’s eye sticker with a central circle of 0.4 cm diameter that served as the fixation point. To ensure that the somatosensory stimuli were inaudible, each participant was provided with headphones over which continuous white noise was played.

The three stimulus conditions were presented randomly with equal frequency and consisted of three blocks of 45 trials (135 trials in total). Anticipatory effects were prevented by utilizing an inter-stimulus-interval that varied randomly from 1–3 s. Each block was separated by a 20-s break in order to reduce fatigue and facilitate concentration. Performance accuracy was defined as the number of accurate stimulus detections divided by 45 trials per condition. Data trimming procedures were purposefully avoided to not bias the distribution of the RT data, and RTs for all inaccurate (i.e., omitted) trials were set to infinity [15]. As in our previous studies, participants with unreliable data (accuracy less than 70% on any one condition (*n* = 43) and extremely long overall RTs (*n* = 3)) were not included [7,16,17].

### 2.2. Clinical Evaluation

As part of the study, individuals participated in a neuropsychological test battery that provided comprehensive assessment of cognitive function, which has been validated in previous longitudinal studies of older adults [18]. Global cognitive function was assessed using the Repeatable Battery for Assessment of Neuropsychological Status (RBANS). The RBANS, a brief cognitive test with several alternate forms that measures five individual cognitive domains (attention, immediate memory, delayed memory, language, and visuospatial skills) as well as a total index of global cognitive functioning, is a well-validated cognitive exam with established test–retest reliability, and published age-appropriate normative data that were used to calculate standardized scores [19].

Cognitive status (normal, mild cognitive impairment (MCI), or dementia) was assessed at each study wave using reliable cut scores from the AD8 Dementia Screening Interview (cutoff score ≥ 2) [20,21] and the Memory Impairment Screen (MIS; cutoff score < 5) [22]. A multidisciplinary team of neurologists, neuropsychologists, and psychologists conducted consensus clinical case conferences where participants’ demographic, neuropsychological, neurological, psychosocial and functional test data were comprehensively reviewed, and diagnosis of cognitive status (normal, MCI or dementia) was assigned annually at each study wave using established criteria [23,24].

During clinical interviews, participants were asked to report the presence or absence of a total of nine physician-diagnosed conditions (including hypertension, arthritis, diabetes, depression, chronic obstructive pulmonary disease, myocardial infarction, angina, stroke, and chronic heart failure) and a total global health score (GHS; range 0–9) was subsequently obtained. Given that the express purpose of the current study was to determine whether presence of diabetes is associated with decreases in VSI processes, which subsequently influence the relationship of VSI with mobility outcomes, presence of diabetes was excluded from the GHS total score (range: 0–8) when used as a covariate in statistical modeling. In addition to diabetes, we also examined the relationship of VSI magnitude with eight other common physician-diagnosed medical comorbidities using bivariate Pearson correlations. After applying Bonferroni correction (*p* < 0.006), results revealed that only presence of diabetes was significantly correlated with decreased magnitude of VSI (r = −0.150, *p* ≤ 0.005).

Of the entire study cohort, 62 individuals (18%) endorsed previous physician-diagnosed diabetes. A study physician (C.G.) confirmed diabetes diagnosis by review of charts, as well as cross-validation of medication lists to determine management of diabetes through oral hypoglycemic agents (OHG) with or without insulin. Of these 62 participants who endorsed presence of diabetes, 14 older adults reported insulin management (with and without other OHG), 4 endorsed that they monitored diabetes with diet only, and the remaining 44 older adults reported prescription of OHG, namely (metformin (*n* = 25), sitagliptin/metformin (*n* = 5), sitagliptin (*n* = 4), glimepiride (*n* = 3), glipizide (*n* = 2), glyburide (*n* = 3), pioglitazone (*n* = 1), or liraglutide (*n* = 1)). Blood tests were available for a subsample of participants (*n* = 248); in support, non-fasting glucose levels of individuals without diabetes (101.54 mg/dL) was significantly lower than non-fasting glucose levels of individuals with diabetes (154.64 mg/dL; *p* < 0.01).

### 2.3. Quantification of Multisensory Integration Using the Race Model Inequality

When two sources of sensory information are presented concurrently, they offer synergistic information that give rise to faster responses [25]. Race models, commonly implemented to examine multisensory effects, are robust probability (P) models that compare the cumulative distribution function (CDF) of combined unisensory visual (V) and unisensory somatosensory (S) reaction times with an upper limit of one (min (P(RT_V_ ≤ t) + P(RT_S_ ≤ t), 1) to the CDF of multisensory visual-somatosensory (VS) reaction times (P(RTVS ≤ t)) [26,27,28]). For any latency t, the race model inequality (RMI) holds when the CDF of the actual multisensory condition (P(RT_VS_ ≤ t)) is less than or equal to the predicted CDF (min (P(RT_V_ ≤ t) + P(RT_S_ ≤ t), 1)). Note that CDFs take all RTs into account. Acceptance of the above RMI suggests that unisensory signals are processed in parallel, such that the fastest unisensory signal produces the actual response (i.e., the “winner” of the race). However, when the actual CDF is greater than the predicted CDF, the RMI is rejected and the RT facilitation is the result of multisensory interactions that allow signals from redundant information to integrate or combine non-linearly.

Methods to calculate the race model violation have been previously described [14]. Figure 2a depicts the group-averaged difference between actual and predicted CDFs (dashed trace) for the entire cohort, where positive values (gray shaded area between 0 and 10th percentile) are indicative of VS integration (i.e., violation of the race model). The RMI was tested using Gondan’s permutation test over the fastest 10% of responses and robust violation was observed (tmax = 14.40, tcrit = 2.08, *p* < 0.001) [29,30]. As in our most recent work [7,16,17], the area-under-the-curve (AUC) during the group averaged violated percentile bins (0–10th) served as the independent measure of magnitude of VS integration. Note, the higher the magnitude, the better (i.e., more efficient) the VS integration ability.

### 2.4. Motor Outcomes

In the current study, measures of mobility included balance and spatial aspects of gait (i.e., pace factor), given their established association with magnitude of VS integration in older adults [7,16,17]. Static balance was assessed using the unipedal stance time test [31,32]. Unipedal stance time is a widely used test of balance: lower scores are associated with neuropathy [31] and predict falls [32] in the elderly. This test was administered twice, and participants’ stance time on one leg for a maximum of 30 s served as the outcome measure. Of the 339 participants, 8 older adults (all without diabetes) were missing the unipedal stance time test, but did complete all other study protocols; therefore, analyses including balance measures are based on a total sample size of 331. All other variables, including covariates, did not contain any additional missing data.

Quantitative gait assessments were conducted on all 339 participants using a 28-foot instrumented walkway with embedded pressure sensors that provide various spatial and temporal gait parameters (GAITRite^®^ from CIR Systems, Franklin, NJ, USA). GAITRite^®^, a valid system for measuring gait performance with excellent test-retest reliability [33,34,35], is widely used in clinical and research settings [36]. Steady-state locomotion was captured over a distance of 20 feet; data from the first and last 4 feet of the instrumented walkway (void of sensors) were purposefully excluded to eliminate initial acceleration and terminal deceleration. Participants were asked to walk on the mat at their “normal walking speed” in a quiet and well-lit room [37]. 

Principal component method was performed on eight individual spatiotemporal gait parameters: gait velocity, stride length, percentage of double support, stride time, stance time, cadence, stride length variability, and swing time variability. The advantage of a factor approach using orthogonal varimax rotation is to reduce a large number of potentially correlated variables (while retaining most of the information) into a smaller number of uncorrelated independent factors that reduces the redundancy across individual variables. We identified a total of three independent gait factors (namely: pace, rhythm, and variability), but given our previous findings [16], only spatial aspects of gait captured under the pace factor were examined. The pace factor score, which loads highly on gait velocity, stride length, and double support percentage (percent of gait cycle with two-feet on the ground), served as the dependent variable in subsequent analyses. 

### 2.5. Statistical Analysis

Data were inspected descriptively and graphically, and the normality of model assumptions was formally tested. Descriptive statistics (M ± SD) were calculated for continuous variables and between group ANOVAs were conducted (see Table 1 below). All data analyses were run using IBM’s Statistical Package for the Social Sciences (SPSS), Version 25. The distribution of maximum unipedal stance time was skewed; therefore, a natural log transformation was applied to achieve normality, and all statistical analyses utilize the transformed value.

Logistic regression analyses were used to determine whether the magnitude of VS integration (the independent variable) could be used to predict presence of diabetes (dichotomous dependent variable). Models were run unadjusted and adjusted for age, gender, education, ethnicity, visual impairment, neuropathy, and global health score (0–8).

In order to determine whether variation in diabetes status (yes or no; dichotomous independent variable) causes variation in magnitude of VS integration (mediator), which in turn causes variation in specific motor outcomes (dependent variables), two separate mediation models were conducted using the SPSS version of Haye’s PROCESS [38]. Mediation analysis demonstrate how a variable’s effect on an outcome can be partitioned into direct and indirect effects that can be quantified using ordinary least squares regression [38]. The first mediation model employed unipedal stance time (natural log transformation) as the dependent variable, while the second model employed pace factor scores as the dependent variable (see Figure 2). 

The direct effect of cognitive impairment on each motor outcome is represented by path c’. The indirect effect of cognitive impairment on each motor outcome through VS integration is the product of path a and path b (ab). The values for each path reported in Figure 2 are equal to the regression coefficients, followed by the corresponding *p*-value. For the indirect effect (ab), the mediation analyses utilize 10,000 bootstrap samples to generate empirically derived representations of the sampling distribution and a 95% bootstrapped confidence interval (CI) [38]. The mediation models were adjusted for age, gender, education, ethnicity, visual impairment, neuropathy, and global health score (0–8).

## 3. Results

Demographic information is presented below in Table 1 for the entire cohort, as well as separated by diabetes status. Compared to individuals without diabetes, the group with diabetes had less participants who were Caucasian, more participants with visual impairments and neuropathy, increased non-fasting glucose levels, decreased magnitude of VS integration, and worse balance and gait performance.

The race model difference waveform for the overall cohort is depicted in Figure 2a (dashed trace). Overall, our results reveal significant and robust VS integration effects (RMI violation or positive values) over the fastest 10% of RTs (highlighted gray box) using an established permutation test [29]. Figure 2b depicts race model difference waveforms by diabetes status, where the solid black trace represents individuals without diabetes, the solid white trace represents individuals with diabetes not on insulin therapy, and the gray dashed trace represents individuals with diabetes on insulin therapy. As shown in Figure 2b, magnitude of area-under-the-curve during the fastest 10% of RTs (highlighted gray box) significantly decreases with increased severity of diabetes.

Results from a fully-adjusted regression analyses are presented below in Table 2 and reveal that magnitude of VS integration is significantly associated with diabetes status (β = −3.21, *p* ≤ 0.01). 

Further, fully adjusted mediation models were investigated to determine whether diabetes causes variation in the magnitude of VS integration, which in turn causes variation in specific motor outcomes. Results from the first mediation model (Figure 3a), which employed unipedal stance as the dependent measure, revealed a significant direct effect (c’; *p* = 0.01) of diabetes on unipedal stance time. However, the indirect effect (ab) of diabetes on unipedal stance time through magnitude of VS integration was significant 95% CI (−0.16, −0.01), as was path a and b. Results from the second mediation model (Figure 3b), which employed pace factor scores as the dependent measure, revealed a significant direct effect (c’; *p* = 0.02) of diabetes on spatial aspects of gait. Additionally, the indirect effect (ab) of diabetes on spatial gait through magnitude of VS integration was also significant 95% CI (−0.09, −0.01), as was path a and b. Collectively, these findings suggest that the presence of diabetes decreases magnitude of VS integration, which in turn directly influences the association of VS integration with both balance and gait performance in older adults.

## 4. Discussion

In the current exploratory study, we demonstrate that decreased magnitude of VS integration is associated with presence of diabetes in older adults. Further, we demonstrate that presence of diabetes significantly influences the magnitude of VS integration, which in turn adversely impacts its association with both balance and gait performance. That is, older adults with diabetes demonstrate significantly reduced VS integration and worse unipedal stance/spatial gait performance compared to individuals without diabetes. 

This finding is directly in line with findings from our most recent study revealing the adverse effect of dementia and mild cognitive impairment (MCI) on the relationship between VSI magnitude and mobility [7]. Particularly noteworthy here is the significant negative association between VS integration and disease severity. Older adults with worse diabetes severity (i.e., on insulin) manifest significantly decreased VS integration compared to participants with less severe diabetes (diet and/or use of oral hypoglycemic agents), and this was associated with poorer motor outcomes. This parallels our findings in older adults with increased cognitive deficits who also manifest significantly decreased VS integration and worse motor outcomes.

The specific structural and functional neuroanatomical correlates responsible for VS integration in older adults with and without diabetes are currently unknown. Recent studies suggest that the PFC serves as the “key driver of flexible multisensory behavior” [39,40]. The PFC also plays a critical role for cognitive and motor processes [41]. The association between cognitive (attention and executive functioning) and motor outcomes (balance [42,43]; gait [44,45,46]; and falls [47,48]) is well-established. Similarly, the association between VS integration and motor outcomes, including balance, falls, and gait and cognitive outcomes, including attention-based performance [7], is also well established. Thus, we hypothesize that sensory, cognitive, and motor processes rely on overlapping neural circuitry involving prefrontal connections to other cortical, subcortical, and cerebellar regions [7].

Research specifically aimed at investigating the interplay of multisensory processes with cognitive and motor outcomes in diabetes is scant. Diabetes is linked to changes in cerebrovascular function, oxidative stress, AGEs, and insulin signaling system impairments [8]. Diabetes is also linked to alterations in cognitive (namely executive function) and motor functioning in older adults. Holtzer and colleagues reveal that diabetes attenuates neural responses in PFC [10]. As well, Ryan et al. report that diabetes was associated with reductions in psychomotor speed [8], which is likely associated with PFC functionality given that the Grooved Pegboard test requires intact sensory, motor, and even cognitive [49] functioning. Collectively, there is compelling evidence to support an adverse effect of diabetes on (multi)sensory, motor, and cognitive functioning.

Surprisingly, multisensory integration was not significantly associated with other common age-related medical co-morbidities, including arthritis, depression, chronic obstructive pulmonary disease, myocardial infarction, angina, stroke, hypertension, or chronic heart failure in our moderately sized sample. 

Overall, our finding that diabetes is associated with decreased magnitude of VS integration, which in turn is associated with worse mobility outcomes, is in line with our aforementioned hypothesis that sensory, cognitive, and motor processes rely on overlapping neural networks, and highlight the need for further studies to determine: (1) if PFC is directly involved with all of three systems, and (2) the impact of diseases, such as diabetes and AD, on these systems. 

### Limitations and Future Directions

This exploratory study is not without limitations. As previously noted, glycated hemoglobin (HbA1c) levels were not part of the parent study and were therefore not available for all study participants. Future studies might consider obtaining fasting glycated hemoglobin levels as an indicator of diabetes control/severity, which may lead to other important scientific discoveries. The current study was cross-sectional; future studies should consider longitudinal design to monitor changes in diabetes severity (and duration) over time and its influence on sensory, cognitive, and motor functioning in aging. Additionally, while our mediation models adjust for chronological age, it is possible that these associations are not specific to only aging cohorts; thus, further in-depth life-span examinations of diabetes (both Type 1 and 2) are clearly warranted.

Future neuroimaging studies are needed to determine the specific structural and functional neuroanatomical correlates responsible for decreased VSI magnitude in older adults with and without diabetes. Specifically, it would be of great interest to determine the impact of vascular diseases, such as diabetes and AD on gray matter volume decreases in neural networks comprising prefrontal regions. We believe that diabetes (much like AD) causes neural disruptions in the PFC, which concurrently affect multisensory, as well as cognitive and motor functionality, due to their reliance on overlapping neural circuitry; however, this notion needs to be tested empirically. 

Lastly, our results underscore the need for mobile and accessible reaction time test of VS integration abilities that could prove instrumental for clinicians and researchers in identifying patients who are at risk for cognitive and functional decline. Such tests could facilitate the development and implementation of individually tailored interventions aimed at ameliorating disability.

## 5. Conclusions

Here we reveal a significant association between magnitude of multisensory integration and diabetes in older adults. In fact, older adults with diabetes demonstrated significantly decreased (i.e., worse) magnitude of VSI compared to older adults without diabetes. Mediation models revealed that presence of diabetes decreases VS integration abilities, which in turn lead to worse balance and gait. Collectively, these findings point towards the possibility of either a vascular or a cerebrovascular disruption, which could be detrimental to older adults, as it is related to a greater risk for falls. Accessible tests of VS integration abilities could prove helpful for clinicians to (1) identify patients who are at risk for falls, (2) aid in remediation/intervention planning, and (3) improve patient outcomes.

## Figures and Tables

**Figure 1 brainsci-11-00285-f001:**
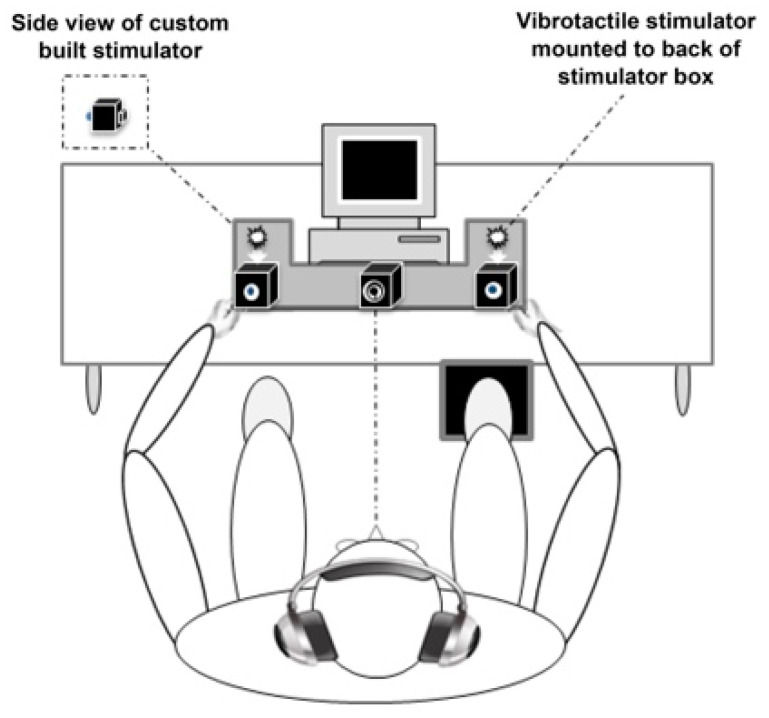
Experimental Apparatus.

**Figure 2 brainsci-11-00285-f002:**
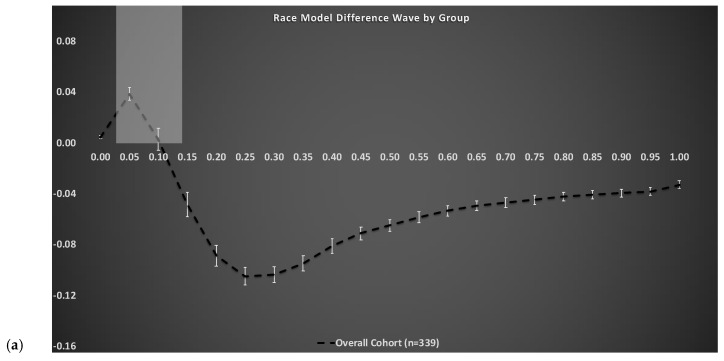
Test of the race model. (**a**) The CDF difference waves over the trajectory of averaged responses for the overall cohort. The grey box highlights the group averaged violated percentile bins (0–10th). (**b**) The CDF difference waves over the trajectory of averaged responses for individuals without diabetes (black trace), individuals with diabetes (white trace), and individuals with diabetes on insulin therapy (dashed trace).

**Figure 3 brainsci-11-00285-f003:**
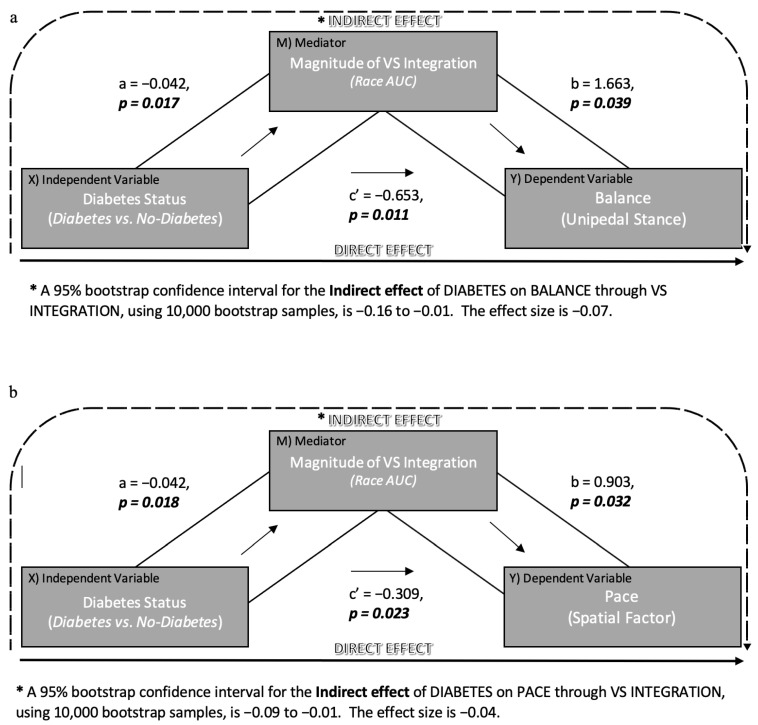
Mediation Analyses. Mediation models along with the results for (**a**) the influence of diabetes on balance through magnitude of VS (visual–somatosensory) integration and (**b**) the influence of diabetes on spatial aspects of gait through magnitude of VS integration AUC: area-under-the-curve.

**Table 1 brainsci-11-00285-t001:** Summary of demographic and clinical variables: overall and split by diabetes status ‡.

Variable			Overall (*n* = 339)	No-Diabetes (*n* = 277)	Diabetes (*n* = 62)	*𝒳* or F Value	* *p*-Value
% Female			52	53	45	1.39	0.24
% Caucasian			74	77	61	**6.42**	**0.01**
% Moderate visual impairment			28	25	39	**4.57**	**0.03**
% Neuropathy			5	3	15	**14.38**	**<0.01**
Age (years)			76.59 (6.21) 65–93	76.71 (6.29) 65–93	76.05 (5.85) 67–92	0.62	0.43
Education (years)			14.97 (2.89) 5–21	15.05 (2.85) 5–21	14.63 (3.09) 7–21	0.51	0.33
GHS Total score (0–8)			1.29 (0.97) 0–5	1.27 (1.00) 0–5	1.35 (0.85) 0–3	2.10	0.50
Glucose ^~^			110.96 (40.17) 68–385	101.54 (20.99) 68–183	154.64 (69.38) 73–385	**75.00**	**<0.01**
VS integration ^#^			0.04 (0.12) −0.33–0.34	0.05 (0.12) −0.32–0.34	0.00 (0.10) −0.33–0.22	**7.26**	**<0.01**
Overall RT (ms)			400.25 (106.10) 243–954	397.26 (104.47) 243–945	413.60 (113.02) 262–954	0.47	0.30
Somatosensory RT (ms)			437.98 (113.05) 252–961	435.64 (113.66) 252–905	448.42 (110.57) 274–961	0.16	0.42
Visual RT (ms)			402.20 (113.77) 233–1050	399.03 (110.98) 233–1050	416.36 (125.44) 250–924	2.66	0.32
VS RT (ms)			361.70 (106.35) 213–1019	358.18 (103.98) 213–1019	377.38 (115.94) 244–975	0.71	0.23
Unipedal stance Time (s)			14.70 (11.09) 0–30	15.67 (11.28) 0–30	10.47 (9.12) 0–30	**20.80**	**<0.01**
PACE	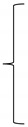	Velocity (cm/s)	99.82 (21.49) 49–167	101.19 (21.51) 49–167	93.70 (20.47) 49–139	**0.77**	**0.01**
		Stride length (cm)	116.61 (19.02) 64–165	117.98 (18.65) 66–165	110.50 (19.59) 64–158	**0.75**	**<0.01**
		Double support (%)	31.61 (4.95) 18–49	31.26 (4.89) 18–48	33.19 (4.95) 24–49	**0.17**	**<0.01**

‡ Values are presented as mean ± SD for continuous variables and % for dichotomous variable. # Area under the curve of the cumulative distribution function (CDF) difference wave over the 0–10% percentile. ~ Non-fasting glucose levels (mg/dL) available for *n* = 248 (204 individuals without diabetes and 44 individuals with diabetes). * Independent samples T-Test or Chi-square (dichotomous variables) results. Abbreviations—GHS: global health score; RT: reaction time; VS: visual–somatosensory.

**Table 2 brainsci-11-00285-t002:** Summary of fully adjusted logistic regression model for predicting diabetes status (diabetes vs. non-diabetes).

Model	B	S.E.	Wald	df	Sig.	Exp (B)	95% CI for Exp (B)	
							Lower	Upper
**VS Integration**	**−3.21**	**1.31**	**5.98**	**1.00**	**0.01**	**0.04**	**0.00**	**0.53**
Overall RT	0.00	0.00	0.03	1.00	0.87	1.00	1.00	1.00
Age	−0.03	0.03	1.27	1.00	0.26	0.97	0.92	1.02
Gender	0.54	0.32	2.85	1.00	0.09	1.72	0.92	3.24
Education Level (years)	−0.03	0.06	0.36	1.00	0.55	0.97	0.87	1.08
Ethnicity	−0.84	0.34	6.28	1.00	0.01	0.43	0.22	0.83
GHS Score ^#^	0.01	0.16	0.00	1.00	0.97	1.01	0.74	1.37
Visual Impairment	−0.69	0.33	4.45	1.00	0.04	0.50	0.26	0.95
Neuropathy	−1.91	0.55	12.25	1.00	0.00	0.15	0.05	0.43

**^#^** GHS score (0–8) does not include presence of diabetes. Abbreviations: B: coefficient; S.E.: Standard Error; Wald: Wald chi-square test; df; degrees of freedom from Wald chi-square test; Sig: significance (*p*-values); and Exp (B): Exponentiation of the B coefficient (i.e., odds ratio).

## Data Availability

Requests for data will be considered upon request.

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
