# Peer review of "The Influence of Diabetes on Multisensory Integration and Mobility in Aging"

_brainsci, 2021, doi:10.3390/brainsci11030285_

Round 1

Reviewer 1 Report

Thank you very much for your invitation to review this interesting study. the topic is important clinically. The authors have investigated the association between diabetes on sensory and mobility impairment in older adults. Overall, the study could be significantly improved.

The main issue is about the study design and findings. the authors should not use the terms “effect” because this in an observational study where it is possible only to identify associations.

The other main concern cross-sectional study where it is difficult to adjust for confounders like other vascular diseases and past medical history. Stroke for example.

Findings from this study is exploratory but conclusion seems to misdirect readers. For example (line 22), “there is compelling evidence to support the adverse effect of diabetes on both multisensory and motor functioning in older adults. This is hard to say, based on the study design and number of patients examined. I do recommend correcting these issues and to include the term of “exploratory findings” in the title.

In line 67, authors stated that “Four-hundred-thirteen participants enrolled in the Central Control of Mobility in Aging (CCMA) longitudinal study”. no information provided on how data collected, who was collecting the data, which year started, and for how long it has been collected. Please provide more details as highlighted.

If the data collected longitudinally, Why authors decided to use cross-sectional study design?

In line 139, the authors mentioned that thy used regression analysis which was not highlighted before (i.e. in the abstract). 

I suggest to provide information on process of variable selection on the  regression model. Is there missing data?

Minor

  1. Table1 and Figure 2 showed findings and presented in the methods section, I suggest moving them into result section.
  2. A box in Figure 2 seems to be shaded. Is that an error?

Author Response

Responses to reviewers’ comment are provided in blue font.

Thank you very much for your invitation to review this interesting study. the topic is important clinically. The authors have investigated the association between diabetes on sensory and mobility impairment in older adults.

Thank you.

Overall, the study could be significantly improved. The main issue is about the study design and findings. The authors should not use the terms “effect” because this in an observational study where it is possible only to identify associations.

I understand the reviewer’s comment/concern here; however, according to Hayes [35] mediation analyses are specifically designed to demonstrate a variable’s effect on an outcome which can be partitioned into direct and indirect effects that are quantified using ordinary least squares regression. Thus, it is accurate to claim that diabetes has an adverse effect on multisensory and motor processes.

The other main concern cross-sectional study where it is difficult to adjust for confounders like other vascular diseases and past medical history. Stroke for example.

This cross-sectional study does take medical co morbidities into account. Our analyses control for presence of: arthritis; depression; chronic obstructive pulmonary disease; myocardial infarction; hypertension; angina; stroke; and chronic heart failure.

Additionally, only 3 participants reported presence of stroke (1 without diabetes and 2 with diabetes). When these three participants are removed from the mediation analyses, the results remain significant, though p-values do actually become even smaller.

Findings from this study is exploratory but conclusion seems to misdirect readers. For example (line 22), “there is compelling evidence to support the adverse effect of diabetes on both multisensory and motor functioning in older adults. This is hard to say, based on the study design and number of patients examined. I do recommend correcting these issues and to include the term of “exploratory findings” in the title.

See response to comment 2 above. Nevertheless, we have revised and include exploratory where appropriate.

In line 67, authors stated that “Four-hundred-thirteen participants enrolled in the Central Control of Mobility in Aging (CCMA) longitudinal study”. no information provided on how data collected, who was collecting the data, which year started, and for how long it has been collected. Please provide more details as highlighted.

Thank you, we have added some important information in the revised manuscript. Also, to save space, we provide references that describe the methodology of the CCMA study in greater detail.

If the data collected longitudinally, Why authors decided to use cross-sectional study design?

The express purpose of this “exploratory” cross-sectional study was to understand which medical co morbidities adversely impact visual-somatosensory integration. We state in the limitation and future directions section of the revised manuscript that future studies should examine the longitudinal effect of diabetes severity and duration over time and its influence on sensory, cognitive, and motor functioning in aging.

In line 139, the authors mentioned that they used regression analysis which was not highlighted before (i.e. in the abstract). 

We have included this detail in the revised abstract.

I suggest to provide information on process of variable selection on the regression model. Is there missing data?

Good point, we have included these details in the methods section of the revised manuscript.

Minor

  1. Table 1 and Figure 2 showed findings and presented in the methods section, I suggest moving them into result section.

Ok, done.

     2.  A box in Figure 2 seems to be shaded. Is that an error?

Yes, that was an error – thank you for pointing that out. We have included a new Figure 3 (old Figure 2).

Thank you for reviewing our manuscript!

Reviewer 2 Report

The present manuscript titled “The influence of diabetes on multisensory integration and mobility in aging” by Mahoney et al. reports whether diabetes interferes in multisensory processes and mobility in aging. I think it is an interesting and well designed study and the results can be of interest of the journal, however I have some questions and comments that should be addressed to sharpen the manuscript.

Major

My only concern about the sample is that authors consider such a wide age range within the same group, assuming that people between 65-93 years old can be part of the same group. Even if the mean ages are equal in both groups, I think age is crucial here and the sample should be subdivided into two different samples. Maybe 65-75 and 76-90, since people are in completely different evolutionary and aging stages. It would be interesting to recalculate considering this separation and check if results are consistent.

Minor

  • Regarding the methods section, the reader would appreciate to have a visual experimental design overview. So, I suggest to include a Figure representing the experimental design.
  • Regarding the results, all the figures and tables which are mentioned in the text from page 8 onwards, are inserted in pages 5-7, which makes no sense and interferes comprehension of the study. Please move all this figures to the results section, and not in the methods section.
  • In addition, please check the table 1, where the numbers have moved upwards in the % moderate visual impairment line.
  • The mediation analysis figure is low quality, authors should submit it in a better resolution.
  • Please revise all the references are OK.

Author Response

Responses to reviewers’ comment are provided in blue font.

The present manuscript titled “The influence of diabetes on multisensory integration and mobility in aging” by Mahoney et al. reports whether diabetes interferes in multisensory processes and mobility in aging. I think it is an interesting and well-designed study and the results can be of interest of the journal, however I have some questions and comments that should be addressed to sharpen the manuscript.

Major

My only concern about the sample is that authors consider such a wide age range within the same group, assuming that people between 65-93 years old can be part of the same group. Even if the mean ages are equal in both groups, I think age is crucial here and the sample should be subdivided into two different samples. Maybe 65-75 and 76-90, since people are in completely different evolutionary and aging stages. It would be interesting to recalculate considering this separation and check if results are consistent.

While we appreciate the suggestion, our analyses adjust for age and as the reviewer has pointed out, there was no significant difference in the mean age or range of ages (similar standard deviations) across = groups. Most medical conditions worsen with age, so to compare 65-75 vs. 76-90 does not seem appropriate here; this is also why we refrain from contrasts between young adults and older adults. 

Minor

  • Regarding the methods section, the reader would appreciate to have a visual experimental design overview. So, I suggest to include a Figure representing the experimental design.

We now include Figure 1, which depicts the experimental apparatus.

  • Regarding the results, all the figures and tables which are mentioned in the text from page 8 onwards, are inserted in pages 5-7, which makes no sense and interferes comprehension of the study. Please move all this figures to the results section, and not in the methods section.

We have placed all figures and tables in more appropriate places when referenced in the text.

  • In addition, please check the table 1, where the numbers have moved upwards in the % moderate visual impairment line.

Thank you, we have revised table 1 accordingly.

  • The mediation analysis figure is low quality, authors should submit it in a better resolution.

We have submitted higher quality images, though it seems as if compressed images were sent to the reviewers in the original manuscript submission.

  • Please revise all the references are OK.

Done

Reviewer 3 Report

The manuscript by Mahoney, Verghese, and George was a relatively comprehensive research on the potential effects of neural circuit changes on the functional outcomes. The authors focused on the diabetes and tested the cognitive and motor functional changes in the patients. The overall study might provide interesting insights in the diagnostic values and involve aging issues. There were some minor-to-moderate concerns:

  • It was quite understood that the medications (Lines 159-166) were considered in the analysis.
  • Statistics results should include F factor together with p value.
  • Due to limited sample size, power analysis should be performed.
  • Section 1. Limitations & Future Directions, the severity of diabetes has been considered. In addition, the duration of diabetic status (time since diagnosis) can be more important. To improve the quality of research, the data should be added into the additional analysis.

Author Response

Responses to reviewers’ comment are provided in blue font.

The manuscript by Mahoney, Verghese, and George was a relatively comprehensive research on the potential effects of neural circuit changes on the functional outcomes. The authors focused on the diabetes and tested the cognitive and motor functional changes in the patients. The overall study might provide interesting insights in the diagnostic values and involve aging issues. There were some minor-to-moderate concerns:

It was quite understood that the medications (Lines 159-166) were considered in the analysis.

Medications are listed in methods section to describe the types of diabetes management regimens employed by our participants. Diabetes medication, or any other medication for that matter, is not controlled for in any of our analyses. Note, the majority of our participants are prescribed several medications due to having at least one medical comorbidity on the average.

Statistics results should include F factor together with p value.

We have included these values where appropriate in Table 1. Regression and mediation models report beta, effect size, or regression coefficients in the tables and figures.

Due to limited sample size, power analysis should be performed.

Reviewer 1 suggested that this study be considered an exploratory study and we have accepted this revision. Our sample is of moderate size (n=339) and is acceptable for an exploratory study. The current sample size provides adequate power (80%) to reduce false negative results (type II error) and this is further evidenced by presence of positive associations. 

Section 1. Limitations & Future Directions, the severity of diabetes has been considered. In addition, the duration of diabetic status (time since diagnosis) can be more important. To improve the quality of research, the data should be added into the additional analysis.

While this exploratory study was designed to specifically examine the effect of diabetes on aging, the parent study was designed to examine central control of mobility mechanisms in aging.  We agree that the duration of diabetes could prove to be a valuable variable here, however, this information was unfortunately not available.  We therefore include this point as a study limitation in our exploratory study.  

Round 2

Reviewer 3 Report

The manuscript entitled by “The Influence of Diabetes on Multisensory Integration and Mobility in Aging” was an interesting research on the potential sensory integration and aging effects in patients with diabetes. The manuscript was improved with updated statistics/methods (such as in Table 1) but remained unsolved concerns. There were:

  • Lines 48-50 has introduced one of the significant aging related neurological disorders, Alzheimer’s disease and it is “to believe that diabetes would be associated with similar declines”; pathological comparisons should be added. For example, does the conditions reported in the current study brains include Abeta deposition or Tau neuropathy ?
  • Lines 159-161, and Lines 362-364 mentioned many diseased conditions; however, due to the study not designed for heart/brain/lung disorders, it is not quite convincing. Furthermore, “hypertension”was not specific, indicating the random effects of the results.
  • Duration of diabetes is very important to help convince. However, the data (such as time from first diagnosis) can not be provided.
  • Lines 316-318, “which in turn directly affects its association with ... in older adults”were unclear. It may also not be specific to aging groups.
  • Useless wording included Lines 217-218 “All other variables including covariates did not contain any additional missing data”.

Author Response

Responses can be found below in blue font.

  • Lines 48-50 has introduced one of the significant aging related neurological disorders, Alzheimer’s disease and it is “to believe that diabetes would be associated with similar declines”; pathological comparisons should be added. For example, does the conditions reported in the current study brains include Abeta deposition or Tau neuropathy ?

The reviewer does raise a significant point. Unfortunately, the current study did not have AB or Tau samples collected. We have included pathological comparisons from the literature in the revised manuscript.

  • Lines 159-161, and Lines 362-364 mentioned many diseased conditions; however, due to the study not designed for heart/brain/lung disorders, it is not quite convincing. Furthermore, “hypertension”was not specific, indicating the random effects of the results.

Hypertension (and other diseases in our comorbidity score) was based on documented physician-diagnosed disease and review of medical records. We have downplayed the random effect of hypertension in the revised submission in light of the Reviewer’s comment. Note that this exploratory study tested the association of VSI with 9 common geriatric diseases, of which only diabetes was significant.  Presence/absence of other physician diagnosed “heart/brain/lung disorders” was not significantly associated with VSI but was nonetheless adjusted for in our statistical models. We can build off these results to examine the association of objective heart/brain/lung disease markers in the future.

  • Duration of diabetes is very important to help convince. However, the data (such as time from first diagnosis) cannot be provided.

We agree with the reviewer however data regarding the onset and duration of diabetes were not consistently available.  We too are curious to know the impact of diabetes duration, severity, and fasting A1C levels on visual-somatosensory integration. As well, we are currently identifying the neural networks associated with visual-somatosensory integration and our future studies aim to determine how neuropathological diseased like diabetes and dementia alter sensory and motor integration. However, this exploratory study is intended to guide future, more in-depth research investigations which are clearly warranted, and we have added this to our limitations section (see also subsequent response).

  • Lines 316-318, “which in turn directly affects its association with ... in older adults” were unclear. It may also not be specific to aging groups.

This line merely conveys the results of the statistical mediation analyses in words.  Here, we are explaining the indirect effect where presence of diabetes influences the magnitude of VSI, which ultimately has an effect on (or influences) balance performance (model 1) and gait (model 2). Our mediation models control for chronological age; though the reviewer is correct – it may not be specific to aging groups. We have clarified this further and have added a line to our limitations section which states:

Additionally, while our mediation models adjust for chronological age, it is possible that these associations are not specific to only aging cohorts; thus, further in-depth life-span examinations of diabetes (both Type 1 and 2) are clearly warranted.

  • Useless wording included Lines 217-218 “All other variables including covariates did not contain any additional missing data”.

This line was included upon Reviewer’s 1 request.  We are happy to remove it if the editor agrees.